# FIB-4 and APRI as Predictive Factors for Short- and Long-Term Survival in Patients with Transjugular Intrahepatic Portosystemic Stent Shunts

**DOI:** 10.3390/biomedicines10051018

**Published:** 2022-04-28

**Authors:** Simone Anna Keimburg, Jens Theysohn, Matthias Buechter, Jassin Rashidi-Alavijeh, Katharina Willuweit, Hannah Schneider, Axel Wetter, Benjamin Maasoumy, Christian Lange, Heiner Wedemeyer, Antoaneta Angelova Markova

**Affiliations:** 1Department of Gastroenterology and Hepatology, University Hospital Essen, Hufelandstrasse 55, 45147 Essen, Germany; simoneannak@aol.com (S.A.K.); matthias.buechter@uk-essen.de (M.B.); jassin.rashidi@uk-essen.de (J.R.-A.); katharina.willuweit@uk-essen.de (K.W.); christian.lange@uk-essen.de (C.L.); wedemeyer.heiner@mh-hannover.de (H.W.); 2Institute of Diagnostic and Interventional Radiology and Neuroradiology, University Hospital Essen, Hufelandstrasse 55, 45147 Essen, Germany; jens.theysohn@uk-essen.de (J.T.); axel.wetter@uk-essen.de (A.W.); 3Department of Gastroenterology, Hepatology and Endocrinology, Hannover Medical School, Carl-Neuberg-Str.1, 30625 Hannover, Germany; schneider.hanna@mh-hannover.de (H.S.); maasoumy.benjamin@mh-hannover.de (B.M.)

**Keywords:** transjugular intrahepatic portosystemic shunt, short- and long-term liver-transplantation-free survival, liver-related event, APRI and FIB-4

## Abstract

(1) Background: Transjugular intrahepatic portosystemic shunt (TIPS) is a standard therapy for portal hypertension. We aimed to explore the association of established baseline scores with TIPS outcomes. (2) Methods: In total, 136 liver cirrhosis patients underwent TIPS insertion, mainly to treat refractory ascites (86%), between January 2016 and December 2019. An external validation cohort of 187 patients was chosen. (3) Results: The majority of the patients were male (62%); the median follow-up was 715 days. The baseline Child—Turcotte–Pugh stage was A in 14%, B in 75% and C in 11%. The patients’ liver-transplant-free (LTF) survival rates after 3, 12 and 24 months were 87%, 72% and 61%, respectively. In the univariate analysis, neither bilirubin, nor the international normalized ratio (INR), nor liver enzymes were associated with survival. However, both the APRI (AST-to-platelet ratio index) and the FIB-4 (fibrosis-4 score) were associated with LTF survival. For patients with FIB-4 > 3.25, the hazard ratio for mortality after 2 years was 3.952 (*p* < 0.0001). Liver-related clinical events were monitored for 24 months. High FIB-4 scores were predictive of liver-related events (HR = 2.404, *p* = 0.001). Similarly, in our validation cohort, LTF survival was correlated with the APRI and FIB-4 scores. (4) Conclusions: Well-established scores that reflect portal hypertension and biochemical disease activity predict long-term outcomes after TIPS and support clinical decisions over TIPS insertion.

## 1. Introduction

Liver cirrhosis is a leading cause of mortality in Western countries [1]. Transjugular intrahepatic portosystemic shunt insertion (TIPS) was first performed in 1988 [2] and has become an established method to reduce portal-hypertension-related complications: studies report an improved control of ascites compared to therapeutic paracentesis in refractory ascites [3], as well as lower rebleeding rates compared to solely endoscopic treatment in variceal bleeding [4]. A better outcome in transplant-free survival has been reported [5] since improvements in the technology of TIPS implantation and the use of covered stents [6]. Additionally, patients on the waiting list for transplantation have lower mortality after TIPS intervention [7]. Most studies have investigated solely short-time survival and few reports are available on long-term follow-up in homogenous patient cohorts.

Since TIPS complications include hepatic encephalopathy, liver function deterioration and cardiac decompensation, proper patient selection and handling is crucial for the achievement of better survival results [8,9]. Several publications discuss prognostic scores for survival prediction according to patient cohort and TIPS indication [10,11]. At present, the model of end-stage liver disease (MELD) score is still the main parameter to predict mortality in patients undergoing TIPS [12,13]. However, MELD does not mirror precisely the grade of portal hypertension and cirrhosis disease severity.

The APRI (AST-to-platelet ratio index) and FIB-4 (fibrosis-4 score) are noninvasive scores that can be used to assess liver fibrosis in patients with chronic liver disease and have been studied extensively in patients with chronic viral hepatitis [14,15] and NASH [16]. The use of APRI and FIB-4 as predictive markers in patients awaiting TIPS has not yet been studied. We investigated the prognostic value of FIB-4 and APRI for the incidence of mortality and liver-related events in a homogenous single-center cohort with liver cirrhosis patients who received TIPS.

## 2. Materials and Methods

All patients with liver cirrhosis without prior liver transplantation, who received TIPS using only covered stents between January 2016 and December 2019, were included. For external validation, a cohort of 187 TIPS patients was used, comprising patients from Hannover Medical School (Hannover, Germany) who received TIPS between January 2009 and December 2019.

Liver cirrhosis was diagnosed based on medical history. All patient information was anonymized according to standards; data were obtained at the point of routine care and were available in the medical records. Demographic characteristics, liver disease etiology, comorbidities and relevant treatment, as well as biochemical and clinical parameters and data from ultrasound and endoscopy studies were analyzed. Patients gave their written informed consent and the study was approved by the local ethics committee (number Nu20-9192-BO) and followed the principles outlined in the Declaration of Helsinki.

Baseline patient characteristics at the time of TIPS implantation were recorded. Liver disease severity was performed using the well-established Child–Turcotte–Pugh (CTP) score, which is based on laboratory tests reflecting liver function (albumin, INR, bilirubin), portal hypertension (ascites) and detoxification (clinical evidence of encephalopathy) [17], as well as model of end-stage liver disease (MELD) score [12,18]. Comorbidity was evaluated by calculating the Charlson–Deyo comorbidity index (CCI) at the time of presentation for TIPS evaluation. The patient cohort was evaluated according to age, with patients divided according to whether they were younger or older than 65 years old. Indication for TIPS insertion was refractory ascites, recurrent variceal hemorrhage, or a combination of both. The TIPS implantation urgency was divided into elective or emergent.

TIPS placement was performed according to standard clinical practice by two experienced interventional radiologists (JT, AW) in one tertiary center, guided by fluoroscopy and ultrasound with the help of a gastroenterology fellow. Covered stent grafts (TIPS Endoprosthesis, Viatorr, GORE, Arizona, AZ, USA) were used in all patients. Portal and hepatic venous pressure were measured invasively using a pressure transducer system and multichannel monitoring. Stent graft was dilatated after a consensus according to measured portosystemic pressure gradient (PSG) before and after TIPS placement. In case of esophageal varices observed on post-TIPS portography, the varices were embolized using Histoacryl and 0.035-inch metal coils (Cook medical, Bloomington, IN, USA). The arterial pressure and the heart rate were monitored noninvasively. The patients were under light or no sedation, except for the emergency TIPS, where unstable patients had assisted ventilation.

Subsequent outpatient care, including a noninvasive ultrasound control as well as laboratory results, was performed on a regular basis according to hospital standards. TIPS malfunction was supposed in the absence of a doppler ultrasound flow, abnormal velocity (<50 cm/s or >200 cm/s), or a significant change in velocity with respect to previous ultrasound examination. In such cases, an invasive hemodynamic study and digital subtraction portography were performed.

Primary endpoint of this study was liver-transplant-free (LTF) survival, as defined from the date of TIPS implantation to either death, loss of follow-up, or date of transplantation. Secondary endpoint was a liver-related event, defined as hepatic decompensation due to persistence of clinically significant portal hypertension despite TIPS implantation, with resulting paracentesis and/or esophageal variceal bleeding, TIPS revision, hepatic encephalopathy grade ≥ 3, or development of hepatocellular carcinoma during follow-up period.

All statistical analyses were performed using SPSS (IBM SPSS Statistics, Versions 27) and GraphPad Prism 9. Continuous variables are presented as median with interquartile range (IQR) and were compared using the Mann–Whitney-U Test for unpaired data. Categorical variables are shown as numbers with percentages and were compared using a chi-squared test or Fisher’s exact test, as appropriate. Survival was analyzed using the logrank test. In order to adjust for potential confounders, multivariate Cox–logistic regression analysis was conducted, including all clinically significant factors tested in the univariate model. In all analyses, *p* < 0.05 was considered as statistically significant.

## 3. Results

### 3.1. Baseline Cohort Characteristics

In total, 84 male (62%) and 52 (38%) female patients with a median age of 57 years were included. The baseline characteristics of the study population are shown in Table 1. The etiology of the liver cirrhosis was alcohol-related in the majority of cases (75%). The baseline CTP stage was A in 14%, B in 75% and C in 11%. A MELD score over 15 points was measured in 32% of the patients. The mean CLIF-C-AD score was 47 (SEM 0.7) and the mean Charlson–Deyo comorbidity index (CCI) was 5 points (21% estimated 10-year survival).

The indication for TIPS insertion was predominantly refractory ascites (86%), while 32% of patients had experienced spontaneous bacterial peritonitis previously and 52% had hepatorenal syndrome in their medical history. In most of the cases in which it was indicated, recurrent gastrointestinal variceal bleeding coiling was additionally performed during the intervention. The patients with esophageal variceal bleeding had lower MELD and CLIF-C-AD scores and better liver synthesis function (measured by protein and albumin), but lower platelet count (90/nL vs. 141/nL, *p* = 0.001).

Patients with hepatic encephalopathy (HE) higher than grade II were excluded and in most cases, no HE was reported (79%).

The TIPS insertion was performed mainly as an elective procedure (134/136 cases). The median PSG before TIPS insertion was 22 mmHg (IQR = 18–25) and after TIPS implantation it was 9 mmHg (IQR = 7–10, no significant difference between both groups). In the majority of cases (68%), the so-called 8 + 2-centimeter covered stent graft with a medium dilation to 8 mm was used. A median PSG reduction of 13 mmHg (IQR = 9–17) was achieved.

Our validation cohort had similar baseline characteristics (Appendix A). In total, 62% of the patients were male and the median age was 59 years. The main etiology of the liver cirrhosis was alcohol-related (112/187, 60%). The baseline CTP stage was A in 3%, B in 87% and C in 10%. The median MELD score was 13 points. The indication for TIPS insertion was therapy refractory ascites. The median baseline PSG was 15 mmHg (IQR = 12–18) and a reduction to a median of 5 mmHg (IQR = 4–7) was achieved post-intervention.

### 3.2. Cohort Follow-Up and Survival

The patients were followed up on a regular basis for a median of 715 days (IQR = 176–1023). The median hospital stay after TIPS implantation was 7 days (IQR = 5–13), irrespective of the TIPS indication. In-hospital mortality occurred in 10 cases; no technical complications were reported. Nine patients were lost to follow-up; seven patients underwent liver transplantation after TIPS intervention (Figure 1A). The patient liver-transplant-free (LTF) survival rates after 1, 3, 6, 12 and 24 months were 92%, 87%, 79%, 72% and 61%, respectively (Figure 1B). The cause of death was cirrhosis-related in 9 out of 47 cases, it was non-hepatic in 21 and no data were available in 17 patients.

### 3.3. Short-Term and Long-Term Survival Predictive Factors

Table 2 describes the differences in the hemodynamic and biochemical parameters between the LTF survival and non-survival groups after 3- and 24-month of follow-up. Overall, a greater PSG decrease after TIPS insertion was associated with LTF survival.

The patients with decreased kidney function, especially those with HRS in their history, showed an increased risk of short- and long-term mortality (Figure 2A, 3-month survival, logrank HR = 2.884, 95% CI = 1.082–7.687, *p* = 0.054; 24-month survival, logrank HR = 2.001, CI = 1.129–3.549, *p* = 0.019).

Sixteen patients died within 3 months of TIPS insertion. A high baseline CRP level (increased CRP level three times the upper limit of normal) was the only discriminative factor significantly associated with negative short-term outcomes (Figure 2B, 3-month survival, logrank HR = 2.765, 95% CI 1.031–7.4316, *p* = 0.048).

By contrast, the long-term mortality after TIPS insertion was not associated with CRP but with decreased platelet count (lower than 100/nl at the timepoint of insertion, Figure 2C, 24-month survival, logrank HR = 1.987, 95% CI 1.069–3.693, *p* = 0.016). Neither bilirubin nor INR or liver enzymes as markers of liver functionality were associated with short- or long-term survival.

However, a combination of markers reflecting portal hypertension and biochemical disease (APRI and FIB-4) was associated with long-term LTF survival. The mortality was significantly higher in patients with APRI ≥ 1 (Figure 2D, 24-month mortality, logrank HR = 2.638, CI 1.463–4.755, *p* = 0.0007). Even more pronounced, mortality was significantly higher in patients with FIB-4 ≥ 3.25 (Figure 2E, 24-month mortality, logrank HR = 3.952, CI 2.225–7.018, *p* < 0.0001).

In order to validate our hypothesis, we examined whether the APRI and FIB-4 were predictive of LTF survival in our external cohort. This confirmed the trend of higher mortality rates in the patients with baseline APRI ≥ 1 (Appendix A, 24-month mortality, logrank HR = 1.779, 95% CI 1.038–3.050, *p* = 0.044), as well as the patients with FIB-4 ≥ 3.25 (Appendix A, 24-month mortality, logrank HR = 1.267, 95% CI 0.701–2.291, *p* = 0.457).

A MELD score of higher than 13 points has been reported to be associated with a higher risk of complications after invasive interventions and lower survival rates [19]. In our cohort, a MELD score of higher than 13 was also predictive of long-term mortality, but not for short-term mortality (Figure 2F, 3-month mortality, logrank HR = 2.656, 95% CI 0.993–7.108, *p* = 0.059; 24-month mortality, logrank HR = 2.051, CI 1.149–3.661, *p* = 0.013).

We adjusted the cohorts for the parameters, which were significant in the univariate analyses and performed multivariate regression analyses. As reported in previous studies, in addition to the MELD score, increased CRP values correlated with increased mortality risk shortly after TIPS insertion (Table 3). However, FIB-4 but not APRI and MELD score, remained an independent factor in long-term mortality (Table 3, logrank HR = 1.389, *p* = 0.001).

### 3.4. Liver-Related Event Incidence during Follow-Up

Additionally, we analyzed the incidence of liver-related events during the follow-up period. The patients who suffered from liver-related complications after TIPS insertion had higher baseline MELD scores and more frequent episodes of SPB before TIPS. The baseline biochemical parameters were not significantly different between the two groups, while the serum protein levels were lower in the group of patients who experienced an event (Table 4).

Similarly, the patients with a FIB-4 ≥ 3.25 experienced significantly more liver-related events 24 months after TIPS insertion (Figure 3E, logrank HR = 2.404, 95% CI 1.434–4.028, *p* = 0.001). The presence of HRS in the history was also a risk factor for events (Figure 3A, logrank HR = 2.121, 95% CI 1.265–3.557, *p* = 0.005). A baseline MELD score greater 13 points was a discriminative factor for event incidence (Figure 3F, logrank HR = 1.755, 95% CI 1.031–2.988, *p* = 0.037).

After the adjustment in the multivariate analysis, the baseline FIB-4 was still associated with an increased risk of experiencing a liver-related event during the follow-up period (Appendix A).

## 4. Discussion

This retrospective observational study followed a homogenous and well-characterized cohort of 136 patients with liver cirrhosis, who received TIPS by experienced interventional radiologists, as well as a validation cohort of 187 patients. We suggest that the simple and well-established scores, FIB-4 and APRI, could be useful to identify patients with a higher risk of experiencing liver-related complications and mortality after TIPS insertion.

Liver-related complications in patients with cirrhosis are associated with high mortality. TIPS insertion reduces portal hypertension and has been shown to improve survival, both in patients receiving TIPS due to variceal bleeding [20] and in those with refractory ascites [5,21]. However, careful pre-selection and the experience of the center are important, since major complications may occur after TIPS insertion, including hepatic encephalopathy, liver function deterioration and/or death [8]. TIPS insertion is not recommended for patients with advanced liver cirrhosis disease, as defined by bilirubin levels > 3 mg/dL, lower platelet count and/or hepatic encephalopathy higher than grade 2 [22].

Several previous studies identified the MELD score and the cause of liver cirrhosis as predictive factors for outcomes after TIPS insertion [12,23,24,25]. In recent years, the use of smaller-caliber stents has led to reduced episodes of hepatic encephalopathy [26] and liver decompensation [27,28]. However, simple prognostic factors predicting short-term as well as long-term mortality after TIPS insertion to guide patient selection are not well established. Here we hypothesized that well-established scores the reflect portal hypertension and biochemical liver disease activity, such as FIB-4 and APRI, could have prognostic value for survival and event incidence after TIPS insertion.

FIB-4 and APRI were validated as non-invasive tools for liver fibrosis assessment in a variety of chronic liver diseases, including viral hepatitis, NASH and alcoholic cirrhosis [29,30,31,32]. Both scores consider platelet levels as markers of portal hypertension and AST and/or ALT levels as indicators of inflammatory disease activity.

In most TIPS cases, short-term mortality results from acute liver function deterioration and hemodynamic decompensation as fragile liver cirrhosis patients lack compensating mechanisms. Our study supports the hypothesis that systemic inflammation is a major risk factor for fatal outcomes in decompensated liver cirrhosis. Elevated C-reactive protein levels over a threshold of 1.5 mg/dL were highly predictive of short-term mortality in a time window of 30–90 days post-intervention, even in the absence of infection or a requirement for antibiotic therapy. Our findings support already published data, which explore the correlation between C-reactive protein, portal hypertension and mortality [33,34,35].

By contrast, the long-term outcomes after TIPS were mainly dependent on portal hypertension markers and liver inflammation, reflected as a combination of markers in the FIB-4 and APRI scores. In a multivariate model, these scores even outperformed MELD as a predictive score for long-term LTF survival. The same tendency was verifiable in our external validation cohort, albeit with some minor differences in the clinical characteristics. The mortality rate among the patients who survived more than 12–24 months was mainly attributed to liver disease deterioration and thus should not be regarded as a direct TIPS complication.

One of the strengths of our study is the relatively large and homogenous cohort, the involvement of the same interventional radiologist and the standard pre- and post-interventional follow-up visits. However, our study also features some limitations. The cohort was a group of selected patients, whose interventions were mainly elective. Thus, the ability of the mentioned scores to predict the long-term outcomes should be evaluated in the setting of emergency TIPS insertion. It is well known that in the case of bleeding and hemodynamic instability, liver enzymes and platelet count can deteriorate quickly and, thus, the value of FIB-4 should be further evaluated in a cohort, which involves more variceal bleeding cases. It should be further noted that our cohort included mainly patients with alcohol-induced liver disease. Here, ongoing alcohol consumption may have contributed to liver deterioration and decompensation events in some patients. Moreover, additional systemic inflammation markers could be helpful to predict short-term survival and have stronger statistical power, since the typical inflammatory markers of liver cirrhosis do not always reflect the actual immune response status. Finally, the findings of this pilot study require validation in independent and larger multicenter cohorts. This task is ongoing.

## 5. Conclusions

In conclusion, we suggest that the simple and well-established scores, FIB-4 and APRI, could be useful to identify patients with a higher risk of experiencing liver-related complications and mortality after TIPS insertion. Furthermore, this study confirms that short-term outcomes after TIPS in patients with significant inflammation are impaired. Thus, TIPS insertion should be considered cautiously in patients with increased inflammatory markers, high biochemical disease activity and severe thrombocytopenia.

## Figures and Tables

**Figure 1 biomedicines-10-01018-f001:**
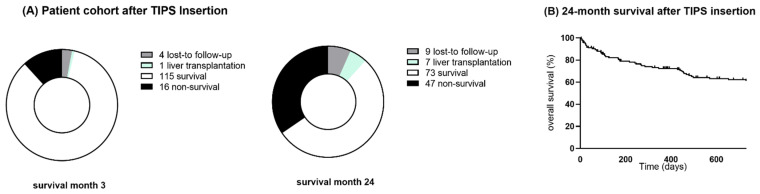
(**A**). Patient cohort 3 months and 24 months after TIPS insertion. (**B**). LTF survival of patients 2 years after TIPS insertion.

**Figure 2 biomedicines-10-01018-f002:**
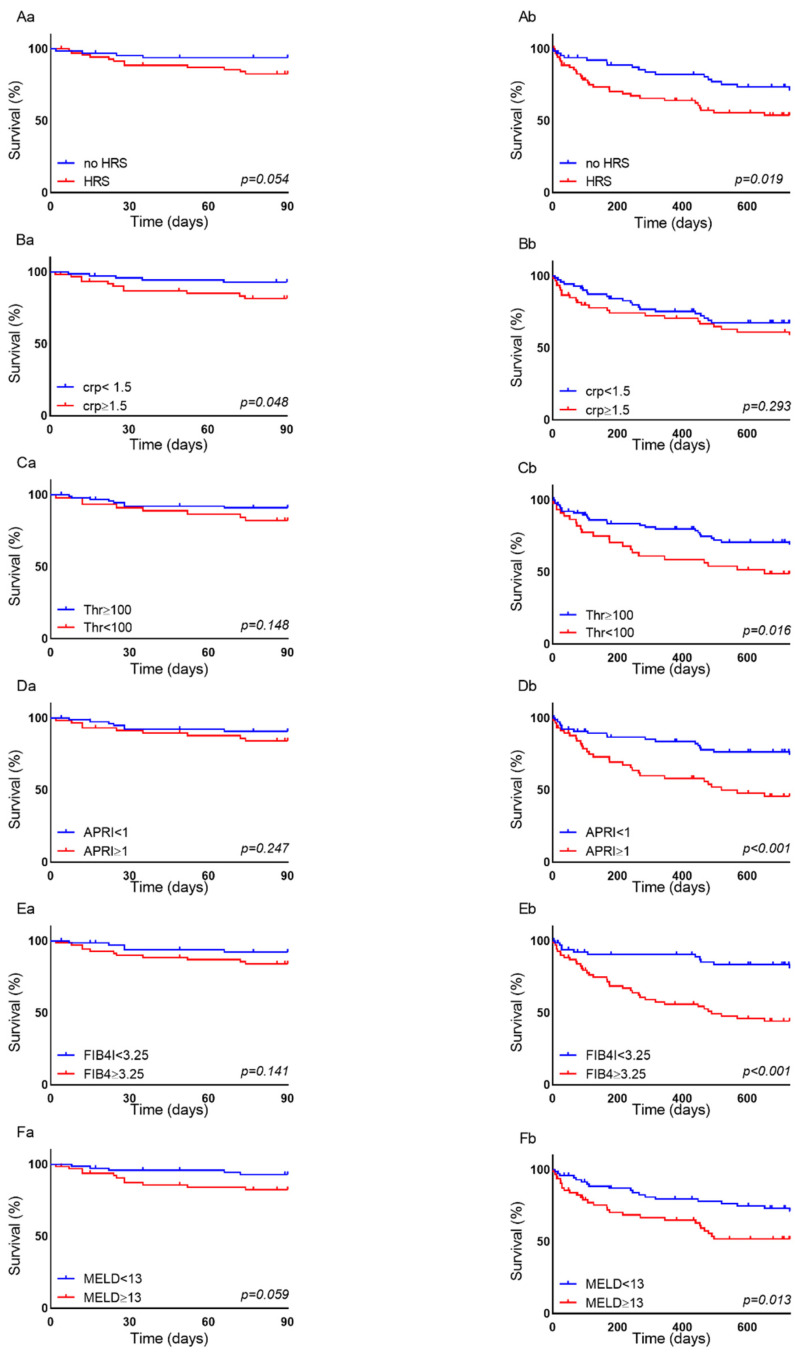
90 days (**a**) and 730 days (**b**) LTF survival of TIPS patients dependent on (**A**) HRS in the history (HRS vs. no HRS), (**B**) CRP levels (<1.5 mg/dL vs. ≥1.5 mg/dL), (**C**) platelet count (≥100/nL vs. <100/nL), (**D**) APRI (<1 vs. ≥1), (**E**) FIB-4 (<3.25 vs. ≥3.25) and (**F**) MELD score (<13 vs. ≥13). The *p*-values were obtained using the logrank test.

**Figure 3 biomedicines-10-01018-f003:**
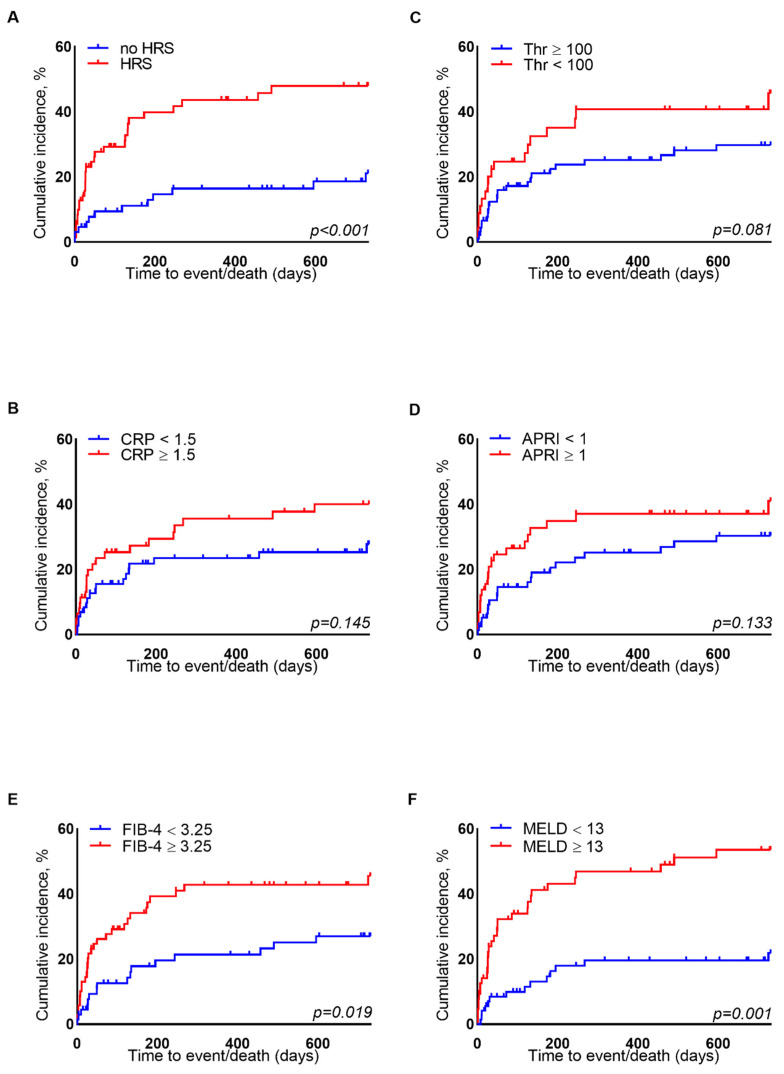
Cumulative event incidence for 730 days after TIPS insertion dependent on (**A**) HRS in the history (HRS vs. no HRS), (**B**) CRP levels (<1.5 mg/dL vs. ≥1.5 mg/dL), (**C**) platelet count (≥100/nL vs. <100/nL), (**D**) APRI (<1 vs. ≥1), (**E**) FIB-4 (<3.25 vs. ≥3.25) and (**F**) MELD score (<13 vs. ≥13). The *p*-values were obtained using the logrank test.

**Table 1 biomedicines-10-01018-t001:** Demographic and clinical characteristics by indication for TIPS at baseline.

	All Patients	Refractory Ascites	Variceal Bleeding	*p*-Value
Patients, n (%)	136 (100)	117 (86)	19 (14)	
Age, y	57 (49–64)	57 (50–64)	54 (37–63)	0.244
Male/female, n (%)	84/52 (62/38)	74/43 (63/37)	10/9 (53/47)	0.377
Etiology of cirrhosis				
Alcohol/viral/autoimmune/NASH/other, n	102/5/9/12/8	93/3/6/9/6	9/2/3/3/2	0.025
Child–Pugh A/B/C, n	19/102/15	6/97/14	13/5/1	0.001
MELD	12 (10–15)	13 (10–16)	11 (8–12)	0.049
CLIF-C-AD Score	47 (42–53)	48 (43–53)	43 (38–48)	0.007
CCI	5 (4–7)	5 (4–7)	4 (3–7)	0.414
BMI, kg/m^2^	25 (22–29)	25 (22–29)	26 (22–33)	0.355
Esophageal varices				
No varices/Grad I-II / III-IV	38/76/22	38/62/17	0/14/5	0.012
GI bleeding, n (%)	52 (38)	33 (28)	19 (100)	<0.0001
HRS yes/no, n (%)	71/65 (52/48)	66/51 (56/44)	5/14 (26/74)	0.015
Previous SPB yes/no, n (%)	44/92 (32/68)	41/76 (35/65)	3/16 (16/84)	0.096
HE, n (%)	29 (21)	23 (20)	6 (32)	0.239
Elective/urgent, n	134/2	117/0	17/2	0.001
TIPS procedural characteristics				
Pre-TIPS PSG, mmHg	22 (18–25)	22 (18–25)	22 (18–25)	0.934
Post-TIPS PSG, mmHg	9 (7–10)	8 (7–10)	9 (7–11)	0.290
Reduction PSG, mmHg	13 (9–17)	13 (9–17)	13 (8–17)	0.719
Stent diameter, mm	8 (6–8)	8 (6–8)	8 (6–8)	0.999
Coil embolization, n (%)	19 (14)	10 (9)	9 (47)	<0.0001
Laboratory characteristics				
Leukocytes, /nL	5.9 (4.1–7.8)	6.1 (4.8–8.5)	3.9 (3.3–5.8)	0.008
Hemoglobin, g/dL	10 (8.6–12)	10 (8.8–12)	9 (8.3–11)	0.336
Platelets, /nL	130 (91–188)	141 (97–219)	90 (62–114)	0.001
INR	1.2 (1.1–1.3)	1.2 (1.1–1.3)	1.2 (1.1–1.3)	0.565
Fibrinogen, mg/dL	277 (201–377)	279 (199–389)	295 (222–311)	0.635
Sodium, mmol/L	137 (134–139)	137 (134–139)	138 (136–141)	0.074
Creatinine, mg/dL	1.2 (1.0–1.6)	1.2 (1.0–1.7)	1.0 (0.9–1.1)	0.002
Bilirubin, mg/dL	1.2 (0.7–1.7)	1.2 (0.7–1.7)	1.3 (0.8–1.7)	0.789
AST, U/L	36 (29–47)	35 (28–42)	46 (34–59)	0.004
ALT, U/L	20 (14–28)	19 (14–25)	30 (18–53)	0.001
Protein, g/dL	6.4 (5.5–7.1)	6.2 (5.5–6.9)	7.1 (6.1–8.0)	0.005
Albumin, g/dL	3.2 (2.8–3.6)	3.1 (2.8–3.5)	3.6 (2.9–4.0)	0.028
CRP, mg/dL	1.6 (0.6–3.1)	1.6 (0.7–3.4)	0.6 (0.5–1.8)	0.029

**Abbreviations:** NASH: non-alcoholic steatohepatitis; MELD: model of end-stage liver disease; CLIF-C AD: chronic liver failure–acute decompensation; CCI: Charlson–Deyo comorbidity index; BMI: body-mass index; GI: gastrointestinal; HRS: hepatorenal syndrome; SPB: spontaneous bacterial peritonitis; HE: hepatic encephalopathy, TIPS: transjugular portosystemic shunt insertion; PSG: portosystemic pressure gradient; INR: international normalized ratio; AST: aspartate aminotransferase; ALT: alanine aminotransferase; CRP: C-reactive protein. The medians with IQR or numbers with percentages are shown.

**Table 2 biomedicines-10-01018-t002:** **A.** Demographic and clinical characteristics at baseline by short-term LTF survival (3 months) and long-term LTF survival (24 months) after TIPS in 131 patients.

	All Patients Month 3	LTF Survival	LTF Non-Survival	*p*-Value	All Patients Month 24	LTF Survival	LTF Non-Survival	*p*-Value
Patients, n (%)	131 (100)	115 (88)	16 (12)		120 (100)	73 (61)	47 (39)	
Age, y	58 (50–64)	58 (50–64)	58 (50–63)	0.712	58 (50–65)	57 (47–64)	60 (53–66)	0.046
Male/female, n (%)	80/51 (61/39)	69/46 (60/40)	11/5 (69/31)	0.592	72/48 (60/40)	42/31 (58/42)	31/16 (66/34)	0.444
MELD	12 (10–15)	12 (10–15)	16 (12–23)	<0.001	12 (10–15)	12 (9–15)	14 (11–17)	0.004
HRS yes/no, n (%)	68/63 (52/48))	56/59 (49/51)	12/4 (75/25)	0.048		34/39 (47/53)	30/17 (64/36)	0.064
Pre-TIPS PSG, mmHg	22 (18–25)	22 (19–26)	18 (14–22)	0.004	22 (18–25)	22 (19–26)	20 (16–23)	0.005
Post-TIPS PSG, mmHg	9 (7–10)	9 (7–10)	7 (6–10)	0.109	8 (7–10)	9 (7–11)	8 (6–10)	0.159
Reduction PSG, mmHg	13 (9–17)	14 (10–17)	11 (7–13)	0.044	13 (10–17)	14 (11–17)	13 (8–16)	0.042
Laboratory characteristics								
Leukocytes, /nL	5.9 (4.0–7.8)	5.8 (3.9–7.8)	6.6 (4.5–8.9)	0.273	5.8 (4.3–8.0)	6.1 (3.9–8.9)	5.6 (4.4–7.4)	0.258
Hemoglobin, g/dL	10 (8.8–12)	10 (8.8–12)	10 (8.3–11)	0.174	10 (8.6–12.0)	10 (9.0–12.0)	9.7 (8.0–12.0)	0.123
Platelets, /nL	130 (91–189)	138 (95–206)	106 (62–150)	0.134	130 (91–186)	144 (98–234)	107 (74–154)	0.001
INR	1.2 (1.1–1.3)	1.2 (1.1–1.3)	1.2 (1.2–1.4)	0.052	1.2 (1.1–1.3)	1.2 (1.1–1.3)	1.2 (1.1–1.3)	0.298
Fibrinogen, mg/dL	279 (201–377)	284 (204–389)	234 (126–312)	0.052	277 (201–377)	298 (216–384)	250 (186–344)	0.229
Sodium, mmol/L	137 (134–139)	137 (134–139)	137 (131–138)	0.235	137 (134–139)	137 (135–139)	137 (134–139)	0.629
Creatinine, mg/dL	1.1 (1.0–1.6)	1.1 (1.0–1.5)	1.3 (1.1–2.2)	0.008	1.2 (1.0–1.6)	1.1 (1.0–1.4)	1.3 (1.0–2.0)	0.009
Bilirubin, mg/dL	1.2 (0.7–1.7)	1.2 (0.7–1.7)	1.4 (0.6–2.5)	0.201	1.2 (0.7–1.7)	1.1 (0.7–1.6)	1.2 (0.7–1.8)	0.723
AST, U/L	35 (29–47)	35 (29–47)	36 (31–48)	0.037	35 (29–47)	34 (27–41)	37 (30–51)	0.051
ALT, U/L	20 (15–28)	20 (15–28)	16 (13–35)	0.051	20 (14–28)	20 (15–26)	19 (14–33)	0.183
Protein, g/dL	6.4 (5.5–7.1)	6.5 (5.7–7.1)	6.0 (5.0–6.4)	0.009	6.3 (5.5–7.0)	6.5 (5.8–7.1)	5.9 (5.3–6.6)	0.002
Albumin, g/dL	3.2 (2.8–3.6)	3.2 (2.8–3.6)	3.1 (2.6–3.5)	0.291	3.2 (2.8–3.6)	3.2 (2.8–3.6)	3.1 (2.9–3.5)	0.535
CRP, mg/dL	1.3 (0.6–2.9)	1.3 (0.6–2.8)	2.1 (0.9–6.6)	0.0002	1.4 (0.6–2.8)	1.3 (0.7–2.7)	1.6 (0.6–3.3)	0.067
APRI	0.8 (0.5–1.4)	0.7 (0.5–1.4)	1.0 (0.6–2.4)	0.034	0.8 (0.5–1.4)	0.7 (0.5–1.1)	1.0 (0.6–1.7)	0.008
FIB-4	3.3 (2.2–5.5)	3.2 (2.1–4.9)	4.7 (3.0–8.2)	0.011	3.3 (2.3–5.5)	2.9 (2.0–4.0)	4.7 (3.3–6.5)	<0.0001

**Abbreviations:** MELD: model of end-stage liver disease; LTF: liver-transplant-free survival; TIPS: transjugular portosystemic shunt insertion; PSG: portosystemic pressure gradient; INR: international normalized ratio; AST: aspartate aminotransferase; ALT: alanine aminotransferase; CRP: C-reactive protein; APRI: AST-to-platelet ratio index; FIB-4: fibrosis-4 score. The medians with IQR or numbers with percentages are shown.

**Table 3 biomedicines-10-01018-t003:** Multivariate Cox regression analyzing risk factors for short- and long-term mortality after TIPS insertion. All statistically significant parameters tested in the univariate analysis were included in the multivariate model.

Risk Factor for Mortality	Multivariate Analysis 3 Months			Multivariate Analysis 24 Months		
	HR	SE	*p*	HR	SE	*p*
MELD	1.228	0.089	0.022	1.094	0.059	0.129
Pre-TIPS PSG	0.839	0.074	0.19	0.889	0.045	0.009
Protein	0.673	0.388	0.307	0.623	0.247	0.060
CRP	1.240	0.100	0.031			
FIB-4	1.225	0.188	0.282	1.389	0.096	0.001

**Abbreviations:** MELD: model of end-stage liver disease; TIPS: transjugular intrahepatic portosystemic shunt; PSG: portosystemic pressure gradient; CRP: C-reactive protein; FIB-4: fibrosis-4 score; HR: hazard ratio; SE: standard error.

**Table 4 biomedicines-10-01018-t004:** Demographic and clinical characteristics at baseline in the groups without and with liver-related events 24 months after TIPS insertion.

	No Event	Event	*p*-Value
Patients, n (%)	77 (57)	59 (43)	
Age, y	54 (47–63)	60 (53–66)	0.001
Male/female, n (%)	45/32 (58/42)	39/20 (66/34)	0.002
Etiology of cirrhosis			
Alcohol/viral/autoimmune/NASH/other, n	60/3/3/6/5	41/2/8/6/2	0.283
Child–Pugh A/B/C, n	14/56/7	5/46/8	0.225
MELD	12 (10–14)	13 (10–16)	0.009
CLIF C AD Score	47 (42–53)	48 (42–52)	0.772
CCI (points)	4 (3–6)	6 (5–8)	0.001
BMI, kg/m^2^	24 (22–29)	25 (22–30)	0.253
Esophageal varices			
No varices/Grade I-II/III-IV	27/36/14	11/40/8	0.045
GI bleeding, n (%)	27 (35)	25 (42)	0.385
HRS yes/no, n (%)	33/44 (43/57)	38/21 (78/22)	0.013
Previous SPB yes/no, n (%)	20/57 (26/74)	38/21 (64/36)	0.069
Elective /urgent, n	77/0	57/2	0.104
TIPS procedural characteristics			
Pre-TIPS PSG, mmHg	22 (19–26)	21 (17–23)	0.004
Post-TIPS PSG, mmHg	9 (7–11)	8 (6–10)	0.061
Reduction PSG, mmHg	14 (10–17)	13 (9–16)	0.055
Stent diameter, mm	8 (6–8)	7 (6–8)	0.065
Coil embolization, n (%)	14 (18)	5 (9)	0.106
Laboratory characteristics			
Leukocytes, /nL	6.0 (4.1–8.8)	5.7 (4.0–7.5)	0.329
Hemoglobin, g/dL	10 (8.9–12)	10 (8.3–12.0)	0.119
Platelets, /nL	150 (104–227)	111 (85–164)	0.009
INR	1.2 (1.1–1.3)	1.2 (1.1–1.3)	0.828
Fibrinogen, mg/dL	289 (203–391)	262 (190–344)	0.275
Sodium, mmol/L	137 (134–140)	137 (134–139)	0.411
Creatinine, mg/dL	1.1 (1.0–1.4)	1.3 (1.1–1.8)	0.004
Bilirubin, mg/dL	1.2 (0.8–1.7)	1.2 (0.7–1.6)	0.372
AST, U/L	36 (30–42)	35 (27–50)	0.319
ALT, U/L	20 (15–27)	18 (14–29)	0.493
Protein, g/dL	6.6 (5.9–7.3)	6.0 (5.3–6.6)	0.0001
Albumin, g/dL	3.2 (2.8–3.6)	3.1 (2.7–3.5)	0.378
CRP, mg/dL	1.4 (0.7–3.0)	1.4 (0.6–3.0)	0.225
APRI	0.7 (0.5–1.1)	1.0 (0.5–1.6)	0.292
FIB-4	2.9 (2.0–3.9)	4.4 (3.0–6.3)	0.0005

**Abbreviations:** NASH: non-alcoholic steatohepatitis; MELD: model of end-stage liver disease; CLIF-C AD: chronic liver failure–acute decompensation; CCI: Charlson–Deyo comorbidity index; BMI: body-mass index; GI: gastrointestinal; HRS: hepatorenal syndrome; SPB: spontaneous bacterial peritonitis; HE: hepatic encephalopathy, TIPS: transjugular portosystemic shunt insertion; PSG: portosystemic pressure gradient; INR: international normalized ratio; AST: aspartate aminotransferase; ALT: alanine aminotransferase; CRP: C-reactive protein; APRI: AST-to-platelet ratio index; FIB-4: fibrosis-4 score. The medians with IQR or numbers with percentages are shown.

## Data Availability

Not applicable.

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
