# Peer review of "FIB-4 and APRI as Predictive Factors for Short- and Long-Term Survival in Patients with Transjugular Intrahepatic Portosystemic Stent Shunts"

_biomedicines, 2022, doi:10.3390/biomedicines10051018_

Round 1

Reviewer 1 Report

This is an interesting study about the patients with TIPS.

It reflects a hard work and may be ussefull in the management of these patients.

It must be published as it is.

Avoid the liver biopsy is general objective in liver disease.

These two biomarkers may be useful as predictive factors for survival in patients with TIPS ,so this manuscript may be interesting for the hematologic community.

Author Response

Dear Reviewer,

Thank you very much for acknowledging the potential importance of the paper.

Best regards

Reviewer 2 Report

The manuscript titled “FIB-4 and APRI as predictive factors for short-and long-term survival in patients with transjugular intrahepatic portosystemic stent shunts” by Keimburg et al., conducted a retrospective study to predict long-term out following trans jugular intra hepatic portosystemic shunt (TIPS) insertion

Abstract:

  1. Child-Turcotte-Pugh stages A, B, and C could be defined
  2. INR, APRI, FIB-4 for that matter, all non-standard acronyms to be defined for the first time as they appear in the text.

Materials and methods:

  1. Patient age rang may be included in the inclusion/exclusion criteria.
  2. Appropriate references or a detailed methodology of assessing liver disease severity score should be included.
  3. Figure labelling could be more simplified in such a way by avoiding multiple sub labelling. For example: “2Aa”

Author Response

The manuscript titled “FIB-4 and APRI as predictive factors for short-and long-term survival in patients with transjugular intrahepatic portosystemic stent shunts” by Keimburg et al., conducted a retrospective study to predict long-term out following transjugular intra hepatic portosystemic shunt (TIPS) insertion

Abstract:

  1. Child-Turcotte-Pugh stages A, B, and C could be defined

Child-Turcotte-Pugh stages are defined as per standard definition. We added the definition of the staging system to the methods section

  1. INR, APRI, FIB-4 for that matter, all non-standard acronyms to be defined for the first time as they appear in the text.

INR, APRI, FIB-4 were defined as suggested in the corrected version of the manuscript.

Materials and methods:

  1. Patient age rang may be included in the inclusion/exclusion criteria. The patient cohort was separately evaluated according to age, being younger or older than 65 years old.

Additional information about age rang was added to the materials and methods part as suggested. “The patient cohort was separately evaluated according to age, being younger or older than 65 years old. “

  1. Appropriate references or a detailed methodology of assessing liver disease severity score should be included.

Liver disease severity was assessed by Child‐Turcotte‐Pugh (CTP) score as well as Model of End‐stage Liver Disease (MELD) score. Appropriate references were added to the materials and methods as suggested (see Manuscript).

  1. Figure labelling could be more simplified in such a way by avoiding multiple sub labelling. For example: “2Aa”

Figure labeling was simplified as appropriate in order to convert the differences between the groups.

Reviewer 3 Report

This study aim to investigate the factors that influence the outcome after TIPS using a cohort from January 2016 until December 2019 and external validation cohort of 187 patients. However, there is some issue required explanation.

1, Please add the details of ascites and portal vein thrombosis in Table 1.

2, How many different types of stents were used? It is better to show the stent diameter by types rather than compared the actual diameters.

3, In the manuscript, "Patients with decreased kidney function, especially with HRS in the history, showed 175 an increased risk for short- and long-term mortality (Figure 2A, 3-month survival, logrank 176 HR=2.884, 95% CI=1.082-7.687, p=0.054; 24-month survival, logrank HR=2.001, CI=1.129- 1773.549, p=0.019).” However, the data of HRS was not found in Table 2.

4, HRS and FIB-4 were found significant in univariate analysis. It should be included in multivariate and explained in the discussion.

5, It would be better to add the comparison between the analysis cohort and the validation cohort.

Author Response

This study aim to investigate the factors that influence the outcome after TIPS using a cohort from January 2016 until December 2019 and external validation cohort of 187 patients. However, there is some issue required explanation.

1, Please add the details of ascites and portal vein thrombosis in Table 1.

117(86%) Patients with refractory ascites were included in the study. Detailed demographic and clinical characteristics of the cohort is available in Table 1. Patients with portal vein thrombosis were not included in the study.

2, How many different types of stents were used? It is better to show the stent diameter by types rather than compared the actual diameters.

Only one type of covered stent grafts (TIPS Endoprosthesis, Viatorr, GORE, Arizona, AZ, USA) were used in all patients. Stent graft was dilatated after a consensus according to measured portosystemic pressure gradient (PSG) before and after TIPS placement. We highlighted this point in the methods.

3, In the manuscript, "Patients with decreased kidney function, especially with HRS in the history, showed 175 an increased risk for short- and long-term mortality (Figure 2A, 3-month survival, logrank 176 HR=2.884, 95% CI=1.082-7.687, p=0.054; 24-month survival, logrank HR=2.001, CI=1.129- 1773.549, p=0.019).” However, the data of HRS was not found in Table 2.

Sorry for this mistake. Data about HRS was added to Table 2A and 2B (see Manuscript attached).

4, HRS and FIB-4 were found significant in univariate analysis. It should be included in multivariate and explained in the discussion.

HRS was not included in the multivariate analysis since it reflects the kidney function and is already reflected in the MELD score. FIB-4 was significant in the multivariate analysis (see Table 3).

5, It would be better to add the comparison between the analysis cohort and the validation cohort.

Both cohorts does not differ significantly in the demographic parameters as shown in Suppl. Table 1. However, as patients were recruited in different centers during different time frames, a more detailed comparison was not performed.